# Effects of Temperature and Water Types on the Decay of Coronavirus: A Review

Ying Guo [1], Yanchen Liu [2], Shuhong Gao [3], Xu Zhou [3], Muttucumaru Sivakumar [1] and Guangming Jiang [1,4,*]

1   School of Civil, Mining, Environmental and Architectural Engineering, University of Wollongong, Wollongong, NSW 2522, Australia
2   State Key Joint Laboratory of Environment Simulation and Pollution Control, School of Environment, Tsinghua University, Beijing 100084, China
3   State Key Laboratory of Urban Water Resource and Environment, School of Civil & Environmental Engineering, Harbin Institute of Technology (Shenzhen), Shenzhen 518055, China
4   Illawarra Health and Medical Research Institute, University of Wollongong, Wollongong, NSW 2522, Australia
*   Correspondence: gjiang@uow.edu.au; Tel.: +61-02-4221-3792

**Abstract:** The analysis of Severe Acute Respiratory Syndrome Coronavirus 2 (SARS-CoV-2) gene copy numbers in wastewater samples can provide quantitative information on Coronavirus Disease-19 (COVID-19) cases within a sewer catchment. However, many wastewater-based epidemiology (WBE) studies have neglected virus decay during the wastewater transportation process in sewers while back-calculating COVID-19 prevalence. Among various sewer condition parameters, wastewater temperature and dilution by fresh/saltwater infiltration may result in a significant change to the virus decay, in terms of both infectivity and Ribonucleic Acid (RNA). This paper reviewed the literature to identify and discuss the effects of temperature and water types (i.e., wastewater, freshwater, and seawater) on coronavirus decay based on the decay rate constants that were collected from published papers. To evaluate the importance of virus decay, a sensitivity analysis was then conducted with decay rates of SARS-CoV-2 RNA based on a WBE back-calculation equation. Finally, the decay rates of coronavirus in wastewater were also compared with those of other viruses to further understand the difference among virus species. The decay of SARS-CoV-2 RNA was found to be less impacted by temperature variation than viable coronaviruses. Nevertheless, WBE back-calculation was still sensitive to the RNA decay rates increased by warm wastewater (i.e., over 26 °C), which could lead to a two-times higher relative variance in estimated COVID-19 prevalence, considering the wastewater temperature variation between 4 and 37 °C in a sewer catchment with a 12-h hydraulic retention time. Comparatively, the sensitivity of the WBE estimation to the enveloped SARS-CoV-2 was greater than nonenveloped enteric viruses, which were less easily degradable in wastewater. In addition, wastewater dilution by stormwater inflow and accompanied cold weather might alleviate the decay of coronavirus infectivity, thus increasing the potential risk of COVID-19 transmission through wastewater. Overall, this paper aims to better understand the impact of in-sewer processes on coronavirus decay and its potential implications for WBE. The outcome could quantitatively inform WBE and improve awareness of the increased risk of COVID-19 infection via wastewater during heavy rainfall events. Given the identified scarcity of data available for coronavirus decay in salt water or with chemical additions, future research on the fate of SARS-CoV-2 subjected to chemical dosing for sewer or wastewater treatment plant operations is recommended.

**Keywords:** wastewater-based epidemiology; virus decay; SARS-CoV-2; sewer; back-calculation; sensitivity analysis

## 1. Introduction

Severe Acute Respiratory Syndrome Coronavirus 2 (SARS-CoV-2) has spread among human communities all over the world in the past three years and remains a threat to

public health and global economy [1]. Providing timely reports on coronavirus disease-19 (COVID-19) cases during outbreaks while using traditional epidemiology approaches is difficult and resource-consuming. Fortunately, wastewater-based epidemiology (WBE) has been demonstrated as an effective method in COVID-19 surveillance at the community level [2,3], since domestic wastewater contains biological information discharged from human bodies, such as loads of bacterial and viral shedding [4]. The detected SARS-CoV-2 concentration in wastewater is proportional to the COVID-19 disease burden in the corresponding catchment [5,6]. By retrospectively linking SARS-CoV-2 concentration in wastewater samples collected at sewer pumping stations or wastewater treatment plants (WWTPs) to virus excretion, WBE can back estimate the prevalence of COVID-19 in populations.

Importantly, WBE back-calculation of COVID-19 cases from SARS-CoV-2 concentration in wastewater needs correction considering the in-sewer virus decay [2]. Failing to include in-sewer decay renders the WBE approach prone to under-predicting the amount of upstream virus shedding. Furthermore, seasonal wastewater temperature variation, seawater intrusion, stormwater inflow, underground water infiltration in sewers, and many other factors [7] can affect the decay kinetics of viruses, thus impacting the infectivity and detectability of SARS-CoV-2 at downstream sampling locations. Wastewater temperature might vary in the range of 4–37 °C depending on the usage in household and weather conditions [2]. Higher temperatures facilitate the virus decay in wastewater compared to lower temperatures [8,9]. During in-sewer transportation, domestic wastewater might be diluted by saline water (i.e., seawater/saline groundwater intrusion, municipal seawater utilization for toilet flushing) or freshwater (i.e., stormwater inflow, or fresh groundwater infiltration), which can have impacts on the decay of viruses in sewers [10]. Correspondingly, the correction factor for WBE back-calculation is the in-sewer decay rate constant, which could be obtained using first-order kinetics [8].

This paper systematically collated the decay rates of SARS-CoV-2 Ribonucleic Acid (RNA) under different water conditions from published literature and evaluated the impact of in-sewer decay on back-calculating COVID-19 cases using sensitivity analysis. In addition, there is still a need to understand the fate of infectious coronaviruses outside the human host after its excretion into wastewater. The decay rates of viral infectivity under various temperatures and water types provide insights into the risk of disease transmission via wastewater exposure, under dynamic weather conditions and sewer environments. Therefore, the decay rates of viable coronaviruses were also collected from published literature through systematic review and compared with the loss of RNA signals in waters.

The SARS-CoV-2 strain, as well as surrogate coronaviruses, were used in the reported decay experiments, being employed as representatives for survivability studies, considering the safety challenges in conducting lab work relating to the highly contagious SARS-CoV-2. The murine hepatitis virus (MHV), transmissible gastroenteritis virus (TGEV), feline infectious peritonitis virus (FIPV), porcine respiratory coronavirus (PRCV), and human coronavirus 229E were accepted as surrogates within the *Coronaviridae* family, given their similarity in size, composition, and morphology [9]. The decay rates of these infectious coronaviruses were included in the collection together with SARS-CoV-2.

Moreover, the persistence of coronaviruses with lipid-containing envelopes might be separate from nonenveloped water-borne viruses that were more commonly investigated in water. To better understand the difference among viral species in environmental fates, the collated decay rates of the coronavirus were further compared to those of norovirus (a widely known nonenveloped enteric virus) and other viruses.

Through systematic review, this study summarized literature data about the decay of coronavirus in municipal wastewater and evaluated the sensitivity of decay rates in WBE back-estimations for COVID-19 prevalence. The results could inform wastewater epidemiology and help with the assessment of public health risks.

## 2. Methods

### 2.1. Literature Search and Data Selection

This study compiled the decay rates of coronaviruses from peer-reviewed papers, which were available online before 21 December 2022. Terms "(coronavirus OR SARS-CoV-2) AND (water OR wastewater OR sewer * OR seawater OR stormwater) AND (survival OR decay OR die-off OR inactivat * OR persistence)" were searched in databases available through Web of Science [9]. Initially, the titles of the search results were screened to decide whether their abstract warranted reading. If the title did not contain sufficient information, the abstract would be further evaluated to determine whether the paper can provide numeric decay rates. The papers that passed the screening through the title and/or abstract were subjected to full-text review. The following inclusion criteria were adopted when reviewing the papers: (i) in English, (ii) having decay experiments in dark conditions, (iii) with justifiable methods for enumeration, e.g., using culture methods or reverse transcription quantitative polymerase chain reaction (RT-qPCR), (iv) presentation of extractable decay rates.

For papers providing decay rate constants ($k$, $d^{-1}$) directly, the calculation approach of $k$ was confirmed to conform with the first-order kinetics as shown in Equation (1), where $C$ is the coronavirus concentration (gene copy/L, or median tissue culture infectious dose, $TCID_{50}$/L) at time $t$ and $C_0$ is the initial concentration. If $k$ was determined using the regression slope of $\log(C/C_0)$ versus time (in days), it would be transformed into $\ln(C/C_0)$. For papers providing $t_{90}$ (i.e., time to achieve 90% reduction in concentration), the times were converted to first-order decay rates by applying Chick's law (Equation (2)) [11]. The fitted decay rate constant was included only if the goodness of fit was acceptable ($R^2 > 0.6$). Meta data including temperature, water types, detection methods, and other publication information were also recorded along with the first-order decay rates (Tables S1 and S2).

$$C = C_0 \times e^{-k \times t}, \tag{1}$$

$$k = 2.303/t_{90}, \tag{2}$$

### 2.2. Data Analysis

The decay rate constants ($k$, $day^{-1}$) were grouped according to the types of water, temperature, and detection methods. The $k$ values were imported into R studio (version 1.3.1056, https://www.rstudio.com (accessed on 26 December 2022) for plotting by gglpot2 package and processed using multcomp for multivariate analysis of variance. When necessary, $k$ values were shown in figures after log10-transformation for better visualization of the decay rate constants over multiple orders of magnitude.

The relationship between temperatures and decay rate constants was fitted according to the Arrhenius equation (Equation (3)), where $k_0$ is the decay rate constant at $T_0$ °C, $k$ is the decay rate constant at temperature $T$ °C, $T_0$ is the reference temperature in °C, $\lambda$ is the temperature correction coefficient indicating the effect of temperature on virus decay rates. The parameters $\lambda$ and $k_0$ were estimated by assuming $T_0 = 0$ °C and fitting Equation (3) to the literature data using SigmaPlot 14 (Systat Software, San Jose, CA, USA).

$$k = k_0 \times \lambda^{(T-T_0)}, \tag{3}$$

The WBE back calculation [2] was adapted as Equation (4), where $F$ is the wastewater flow rate (L/day), $C_{RNA}$ is the concentration of SARS-CoV-2 RNA in wastewater (copies/L), $P_{catchment}$ is the population within the sampled catchment boundary, $E$ is the daily excretion rate (copies/(person·day)). To account for the in-sewer decay, a correction factor $R_{sewer}$ was applied. As the $C_{RNA}$ in Equation (4) should be $C_t$ after decay, $R_{sewer}$ can be derived from the decay kinetics as $C_t/C_0$ (refers to $e^{-k \times t}$ as in Equation (5)). The number of infected cases $P_{infection}$ thus can be corrected using the decay factor $R_{sewer}$, where $t$ denotes in-sewer

hydraulic retention time during wastewater transportation and $k$ refers to the decay rate constant, as described above.

$$P_{\text{infection}} = (F \times C_{\text{RNA}})/(P_{\text{catchment}} \times R_{\text{sewer}} \times E), \tag{4}$$

$$R_{\text{sewer}} = C_t/C_0 = e^{-k \times t}, \tag{5}$$

The sensitivity ($S$) of WBE-calculated infection case number ($P_{\text{infection}}$) to the decay rate constant ($k$) of SARS-CoV-2 RNA is defined in Equation (6). The relative change of $P$ ($\Delta P/P$), due to the change of $k$ by $\Delta k$, can be deduced as $\exp(\Delta k \times t) - 1$. The average $k$ values from the collected literature data were taken as inputs into Equation (6), assuming the relative change as 25% of the decay rate constants [10]. The hydraulic residence time in sewer systems was assumed to be 12 h, representing a typical large sewer catchment. Herein, the calculated sensitivity value reflects the percentage of variation in WBE back-calculation caused by the variation of parameter $k$.

$$S = (\Delta P/P)/(\Delta k/k) = k \times (e^{t \times \Delta k} - 1)/\Delta k, \tag{6}$$

## 3. Results and Discussion

A total of 79 decay tests for coronavirus in water were identified from 12 unique research articles (Table 1, Tables S1 and S2). One decay test refers to an in-water decay experiment at a certain temperature in one type of water, either for virus infectivity or RNA. These studies employed different temperatures for the decay experiments in water, and the types of water included wastewater, freshwater, and seawater. Wastewater denotes domestic or municipal wastewater, including sterilized, autoclaved, and untreated raw wastewater. Freshwater includes tap water, laboratory water, and river water. The decay rate constant ($k$) was presented as $\ln(C/C_0)$ per day. Notably, there is just one decay rate in seawater measured by RT-qPCR (0.14 with $R^2$ as 0.8, at 20 °C) that met the selection criteria, which was reported by Sala-Comorera et al. [12]. Moreover, the numbers of infectivity decay in seawater and SARS-CoV-2 RNA decay in freshwater are limited, with only three for each (Table 1).

**Table 1.** Summary of identified decay tests of coronavirus in different water types.

| Methods | Culture | RT-qPCR | Total |
|---|---|---|---|
| Number of decay tests in wastewater | 18 | 33 | 51 |
| Number of decay tests in freshwater | 21 | 3 | 24 |
| Number of decay tests in saline water | 3 | 1 | 4 |
| Total | 42 | 37 | 79 |
| References | [12–18] | [14,19–23] | |

### 3.1. Overview of Coronavirus Decay Rates in Waters

Table 2 summarizes the collected decay rates from the reviewed papers under different conditions and Table 3 shows the result of the multivariate analysis of variance. The collected dataset contained two category variables (detection methods, type of water) and water temperature as a continuous variable. Multivariate analysis of variance showed that all these three variables led to significant differences in decay rate constants ($p < 0.001$) (Table 3). Most significantly, the collected decay rates were impacted by the different detection methods ($p = 1.68 \times 10^{-8}$). The three SARS-CoV-2 RNA decay rates (0.96–4.32 d$^{-1}$) reported by Weidhaas et al. [19] were significantly higher than average reports, thus being identified as outliers and excluded from the following modeling analysis. After the exclusion, the decay rate constants of SARS-CoV-2 RNA in wastewater ranged from 0.04 to 0.7 for temperatures 4–37 °C (Table 2), whereas the infectivity of coronaviruses decays at much faster rates (0.066–3.4 d$^{-1}$), from 4 to 25 °C in wastewater. In freshwater, the decay rates of SARS-CoV-2 RNA (0.039–0.25 d$^{-1}$ at 4–37 °C) were also lower than those of coronavirus infectivity (0.01–1.2 d$^{-1}$ at 4–24 °C), although the difference between two methods was not as much as that in wastewater.

The nucleic acids of viruses may persist longer than viral capsid and remain detectable after losing infectivity; hence, the decay rates of SARS-CoV-2 RNA are supposed to be much lower than those of coronavirus infectivity [14]. The greater stability of genetic fragments makes them suitable candidates for WBE investigations. At the same time, it also indicates that the environmental detection of viral RNA alone does not substantiate the risk of infection. On the other hand, knowledge about the fate of infectious viruses is needed to evaluate the potential disease transmission through urban wastewater systems, especially through sewers, where stormwater infiltration and saltwater intrusion might occur. Considering the scarcity of available data for seawater, the subsequent analysis and discussion mainly focused on wastewater dilution via freshwater inflow.

**Table 2.** Summary of decay rate constants ($d^{-1}$).

| Target | Water Types | 4 °C | 10–15 °C | 20–26 °C | 37 °C |
|---|---|---|---|---|---|
| Coronavirus Infectivity | Wastewater | 0.066–0.42 | 0.5–1.4 | 0.6–3.4 | NA |
| | Freshwater | 0.01–0.61 | NA | 0.2–1.2 | NA |
| | Seawater | 1.1 | NA | 2.0–2.1 | NA |
| SARS-CoV-2 RNA | Wastewater | 0.04–0.18 | 0.08–0.4 | 0.17–0.70 | 0.29–0.41 |
| | Freshwater | 0.039 | NA | 0.15 | 0.25 |
| | Seawater | NA | NA | 0.14 | NA |

Note: NA means not available.

**Table 3.** Summary of multivariate analysis of variance.

| | Df | Sum Sq | Mean Sq | F Value | $p$ (>F) | |
|---|---|---|---|---|---|---|
| Water type | 2 | 7.248 | 3.624 | 9.332 | 0.000262 | *** |
| Method | 1 | 15.904 | 15.904 | 40.954 | $1.68 \times 10^{-8}$ | *** |
| Temperature | 1 | 12.485 | 12.485 | 32.151 | $3.17 \times 10^{-7}$ | *** |
| Water type: method | 1 | 0.495 | 0.495 | 1.275 | 0.262756 | |
| Water type: temperature | 2 | 1.162 | 0.581 | 1.496 | 0.231343 | |
| Method: temperature | 1 | 4.781 | 4.781 | 12.311 | 0.000804 | *** |
| Water type: method: temperature | 1 | 1.098 | 1.098 | 2.826 | 0.097312 | |
| Residuals | 68 | 26.407 | 0.388 | | | |

Note: *** means significant as $p \leq 0.001$.

### 3.2. Effect of Wastewater Dilution on Coronavirus Decay

The decay rates of SARS-CoV-2 RNA in seawater (0.14 $d^{-1}$ at 20 °C) and in freshwater (0.039 $d^{-1}$ at 4 °C, 0.15 $d^{-1}$ at 25 °C, 0.25 $d^{-1}$ at 37 °C) were smaller than those in wastewater (0.04–0.18 $d^{-1}$ at 4 °C, 0.17–0.70 $d^{-1}$ at 20–26 °C, 0.29–0.41 $d^{-1}$ at 37 °C) (Table 2). Thus, the freshwater infiltration and seawater intrusion could alleviate the decay of viral RNA and may alleviate the impact of decay on WBE back-calculations.

On the contrary, the reduction in virus viability was enhanced in seawater, which led to higher decay rates (1.1 $d^{-1}$ at 4 °C and 2.0–2.1 $d^{-1}$ at 20 °C) than in wastewater (0.066–0.42 $d^{-1}$ at 4 °C and 0.6–1.4 $d^{-1}$ at 20 °C). However, such a difference is less confirmative, based on the limited data points. Sun et al. [24] suggested that the virus infectivity was not significantly affected by seawater; however, Lee et al. [25] reported a rapid infectivity loss of SARS-CoV-2 immediately upon being introduced into seawater. It should be noted that Sun et al. [24] examined the stability of the virus in artificial seawater while Lee et al. [25] used real seawater (pH 8, salinity 32‰; Sokcho, Korea). The contradictory results might be caused by the indigenous microbial communities in the marine ecosystem, which might inactivate viruses via proteolytic or nuclease activity [26].

Generally, the salinity and alkalinity of seawater are believed to influence the osmotic pressure and may reduce the survivability of coronaviruses in the ocean, thus helping to eliminate the risk of virus transmission [13,27,28]. Although SARS-CoV-2 is a mammalian virus, the release of viable coronaviruses through municipal sewage into the ocean may affect the marine ecosystem and even cause virion accumulation in seafood such as oysters or fishes through food chains in the same way as norovirus [29]. Interestingly, the COVID-19 outbreak first originated from a seafood market in Wuhan. Later, many infection cases

caused by frozen cold chain food products have been reported around the world [30,31]. More consolidated experimental evidence should be involved to confidently exclude the possibility of marine contamination and confirm the safety of the seawater environment even for recreation purposes. Moreover, more experimental investigations in seawater mixed with domestic wastewater are necessary to delineate the effects of salinity on SARS-CoV-2 decay rate constants [32].

Figure 1 illustrates that both coronavirus infectivity and SARS-CoV-2 RNA decay more rapidly in wastewater than in freshwater under different temperatures. The difference at a lower temperature (4 °C) may not be as significant as that at higher temperatures (>20 °C). The increased $k$ of coronaviruses in wastewater compared to freshwater could be attributed to the deactivation from higher extracellular enzymatic activity, eukaryotic predation, or the presence of antiviral chemicals (such as solvents, detergents, etc.), and other organic matters in wastewater [7]. Unlike neutral freshwater, wastewater usually has an acidic pH with fatty acids and other constituents that affect virus decay through the denaturation of capsid proteins and damage of nucleic acids [2]. Wastewater dilution via stormwater could alleviate the inactivation of infectious coronaviruses, thus potentially leading to the release of viable viruses into natural water bodies or city catchments via sewer overflow or untreated discharges [33–35]. Additionally, storm weather is often accompanied by lower temperatures and shorter in-sewer hydraulic retention times, which favor virus survival in waters. Therefore, the risk of spreading infective viruses during heavy rainfall events and urban floods should be carefully evaluated given the ever-increasing virus load in municipal wastewater.

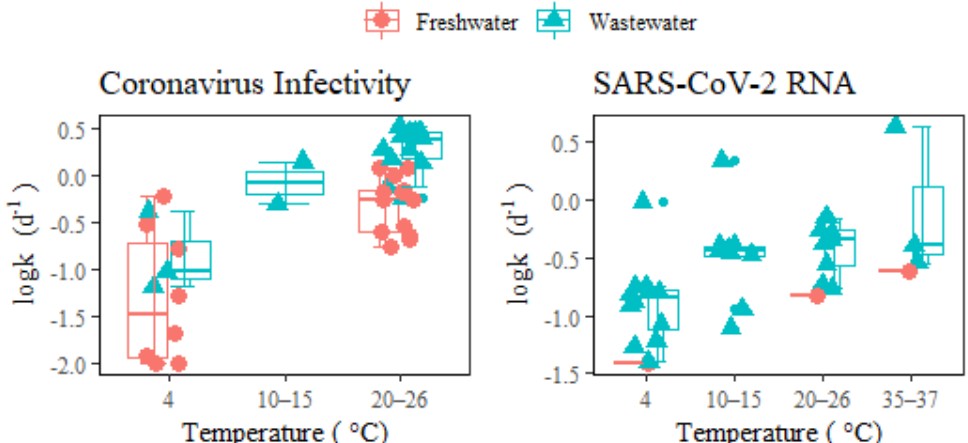

**Figure 1.** Decay of coronavirus infectivity and SARS-CoV-2 RNA in freshwater and wastewater under different temperatures. The middle lines inside the box represent median $k$ values. The top and bottom borders of the box represent the 75%ile and 25%ile of $k$ values, respectively. The top and bottom whiskers represent the upper and lower limit of $k$ values in the groups. In addition, the smaller dots represent the outliers.

### 3.3. Effect of Temperature on Coronavirus Decay

Figure 2 shows the fitted curves for the numerical relationship between decay rate constants and temperatures in wastewater and freshwater based on the Arrhenius equation (Global goodness of fit: $R^2 = 0.84$). Separate plots for the four curves are presented in Figure S1. Table 4 shows the estimated parameters of the temperature correction coefficient ($\lambda$), decay rate constant ($k_{20}$) at a reference temperature of 20 °C, and $R^2$ for each scenario. The poor fitness and wide confidence band (Table 4 and Figure S1) for coronavirus infectivity decay in freshwater might be due to biological differences within coronavirus surrogates and the variability in matrix conditions (tap water versus river water). The even lower fit of SARS-CoV-2 RNA decay in wastewater ($R^2 = 0.2$) could be a result of the varied detection assays applied for RT-qPCR detection (Table S2) or the wastewater conditions (autoclaved, sterilized, or untreated wastewater sampled from various locations), which made the comparison among

fitted parameters $\lambda$ and $k_{20}$ less convincing. More studies via well-controlled experiments are needed to obtain consistent and reliable conclusions. The detailed reporting of environmental parameters for decay tests should also be encouraged to help understand the influencing factors across different studies.

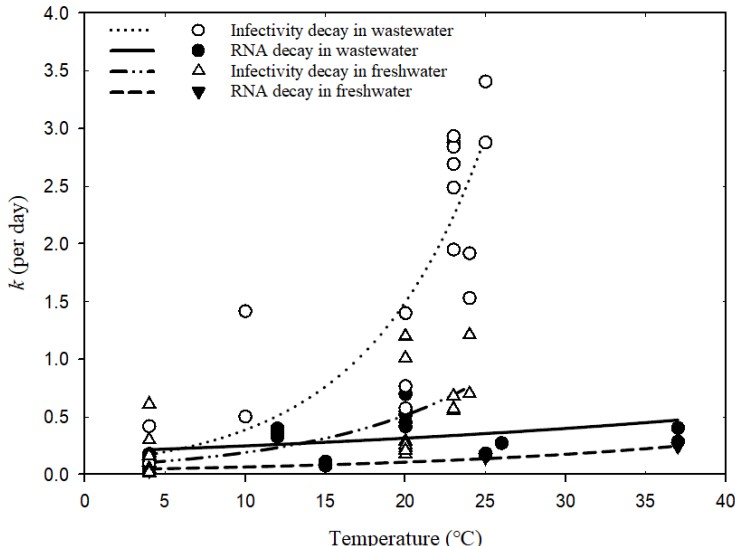

**Figure 2.** Relationships between decay rate constants of coronavirus and temperatures based on Arrhenius equation (Equation (3)).

**Table 4.** Parameters in Arrhenius equation fitted by temperatures and decay rates.

| Parameters | Water Type | Coronavirus Infectivity | SARS-CoV-2 RNA |
|---|---|---|---|
| $k_{20}$ | Wastewater | $1.37 \pm 0.10$ | $0.30 \pm 0.04$ |
| | Freshwater | $0.47 \pm 0.07$ | $0.11 \pm 0.01$ |
| $\lambda$ | Wastewater | $1.14 \pm 0.05$ | $1.02 \pm 0.01$ |
| | Freshwater | $1.10 \pm 0.05$ | $1.05 \pm 0.01$ |
| $R^2$ | Wastewater | 0.72 | 0.20 |
| | Freshwater | 0.42 | 0.99 |

According to the estimated $k_{20}$ values in Table 4, the infectivity decay rates were much higher than SARS-CoV-2 RNA, which indicates a greater stability of RNA than the hosting virus body in the water environment [26] and resonates with the previous discussion in Section 3.1. Moreover, both viable coronaviruses and SARS-CoV-2 RNA presented higher stability in freshwater over the wastewater matrix, probably due to the lower abundance of microbial communities and limited biological activity in freshwater, where degradation is alleviated for viruses [36]. This also supports the comparisons made in Section 3.2.

For each group in different water types, the temperature correction coefficient $\lambda$ was calculated to be greater than one, indicating the influential role of water temperature on decay rates. Higher temperatures would enhance virus decay since the decay rate constants increase with elevated temperatures (Figure 2). In particular, the infectivity decay of coronavirus in wastewater was most significantly enhanced by temperature increase ($\lambda = 1.14 \pm 0.05$), followed by the infectivity decay in freshwater ($\lambda = 1.10 \pm 0.05$). Warmer waters normally increase decay rates of microorganisms via activating enzymes and, consequently, the degradation of protein walls or nucleic acids [37]. Exposure of viruses to high temperatures may also inactivate the enzymes and prevent replication. Since wastewater has more biologically and chemically active components than freshwater, the decay of viable viruses would be more vulnerable to temperature variation in wastewater.

For less sensitive SARS-CoV-2 RNA, temperature increments seem to play a similar role in facilitating decay both in freshwater and wastewater, as suggested by the comparable values of $\lambda$ (Table 4) and the two visually parallel curves in Figure 2. Nucleic acid fragments inside the viral particles are less easily degradable by extracellular enzymes and more persistent than virus infectivity [14,38]. The genetic sequence may degrade gradually into smaller pieces and thus allow the RNA fragments to be prolonged after losing the protection from viral capsid. Hence, their decay rates are less impacted by temperature elevation than those of viable viruses, regardless if in freshwater or wastewater. Although the change in SARS-CoV-2 RNA decay rates in wastewater caused by seasonal temperature variation was much less than those of viable viruses, further assessment regarding its impact on WBE back-estimation should still be useful for understanding the resulting accuracy implications.

### 3.4. Sensitivity of WBE Back-Calculation to the Decay of SARS-CoV-2

To quantitatively assess the contribution of SARS-CoV-2 RNA decay under varied temperatures to the WBE back-calculation accuracy, the collected $k$ values of SARS-CoV-2 RNA were input to the proposed sensitivity equation (Equation (6)) and the calculated results were presented in Table 5. The average level of sensitivity to 25% change in $k$ is 0.13 over the three water types at 4–37 °C. Wastewater dilution by freshwater or saltwater in sewers might result in less sensitivity; however, the lack of data for SARS-CoV-2 RNA in saltwater made any solid conclusions hard to reach in the current study. More confidently, higher temperatures could increase SARS-CoV-2 RNA decay in sewerage and, correspondingly, the uncertainty of WBE back-estimation. Increased wastewater temperature in summer or tropical regions from 4 to 37 °C could increase the sensitivity considerably from 0.06 to 0.18, leading to a two-fold larger relative variance in the estimation of COVID-19 cases.

**Table 5.** Sensitivity ratio of predicted COVID-19 cases ($P_{\text{infection}}$) to SARS-CoV-2 RNA decay rate constants under different conditions assuming residence time in sewers as 12 h.

| Temperature | Wastewater | Freshwater | Sea Water | Average |
|:---:|:---:|:---:|:---:|:---:|
| 4 °C | 0.06 | 0.02 | NA | 0.06 |
| 12–15 °C | 0.15 | NA | NA | 0.15 |
| 20–26 °C | 0.23 | 0.08 | 0.07 | 0.21 |
| 37 °C | 0.18 | 0.12 | NA | 0.16 |
| Average | 0.14 | 0.07 | 0.07 | 0.13 |

Note: NA means not available. The sensitivity is represented as the ratio between the percentage change of estimated COVID-19 cases and 25% change of decay rate constants.

While the case number would possibly be underestimated in hot seasons if not incorporating RNA decay rates in the back-calculation, WBE back-calculation could assess disease prevalence more closely based on wastewater samples with lower temperatures, whether in winter or cold countries, where the importance of considering in-sewer RNA decay is reduced. Since the sensitivity values in Table 5 were calculated with hydraulic retention time fixed as 12 h, the overall sensitivity will be decreased, given a shorter in-sewer travel time, or increased otherwise. Collectively speaking, the in-sewer decay of SARS-CoV-2 RNA could be an influential factor for WBE back-calculation under high environmental temperatures and should be considered for accurate prediction.

### 3.5. Comparison of Coronavirus Decay Rates in Wastewater to Norovirus and Other Viruses

To put coronavirus decay rates into a broader context, Table 6 compared them with other previously investigated viruses (i.e., noroviruses, Zika virus, Dengue virus, yellow fever virus, Ebolavirus, Human Immunodeficiency Virus, hepatitis A virus, adenovirus). This table was based on a previous compilation [39] with the addition of recent reports. Although SARS-CoV-2 RNA might decay slightly faster at 4–6 °C, its overall decay rate was quite similar to other enveloped single-stranded RNA viruses (Zika, Dengue, and yellow fever virus), and generally

ranged from 0 to 0.89 d$^{-1}$ at higher temperatures. As for infectivity, the coronavirus also experienced similar decay rates at room temperatures as other enveloped single-stranded RNA viruses (Ebolavirus and Human Immunodeficiency Virus).

**Table 6.** Decay rate constants $k$ (d$^{-1}$) of different viruses in wastewater.

| Virus | Method | 4–6 °C | 10–25 °C | 30–37 °C | References |
|---|---|---|---|---|---|
| Norovirus | RT-qPCR | 0.02–0.06 | 0.02–0.10 | 0.05–0.21 | [40,41] |
| Zika | RT-qPCR | 0.025–0.046 | 0.11–0.58 | 0.27–0.89 | [42] |
| Dengue | RT-qPCR | 0.008–0.052 | 0.50–0.55 | 0.55–0.61 | [43] |
| Yellow fever | RT-qPCR | 0.032–0.047 | 0.52 | 0.88 | |
| Murine hepatitis virus | RT-qPCR | 0–0.003 | 0.37 | 0.45 | |
| Coronavirus | RT-qPCR | 0.04–0.18 | 0.08–0.70 | 0.29–0.41 | [12–23] |
| | culture | 0.066–0.42 | 0.5–3.4 | | |
| Ebolavirus | culture | | 0.35–1.08 | | [44] |
| Human Immunodeficiency Virus | culture | | 0.80 | | [45] |
| Hepatitis A | culture | 0.047–0.066 | 0.10–0.28 | 0.34 | [46] |
| Adenovirus | culture | 0.10 | 0.10 | | [47] |

It was commonly hypothesized that the lipid-containing envelope that surrounds the coronavirus nucleocapsid made it more susceptible to degradation than nonenveloped enteric viruses (i.e., hepatitis A, adenovirus, and norovirus). The inactivation process for enveloped viruses was observed to be faster than for nonenveloped viruses [18,48], since envelope lipids could be more easily destroyed than other viral parts. As an example, the hepatitis A virus had better survivability than the enveloped coronavirus at 4–25 °C (Table 6). In another instance, the RNA decay rates of a typical nonenveloped enteric waterborne norovirus [10] were significantly lower than coronavirus in wastewater (Figure 3). Such a conclusion contrasts with the opinion of Silverman and Boehm [9], who identified similar persistence between nonenveloped and enveloped viruses in a dark aqueous environment. This discrepancy might be due to the fact that the study combined different water matrices (including wastewaters and natural waters, i.e., fresh, estuarine, and marine waters) together in the comparison. The different water matrices should be considered in view of the significant effect from water types on virus decay (Table 3).

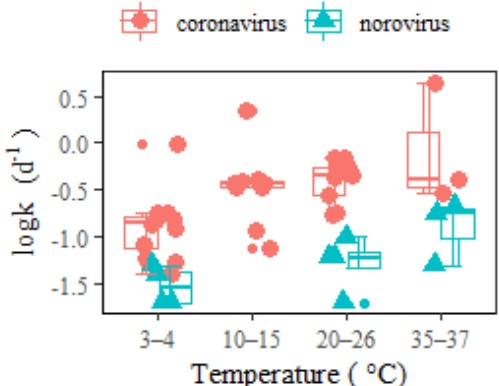

**Figure 3.** RNA decay rates of coronavirus and norovirus in wastewater at different temperatures. The middle lines inside the box represent median $k$ values. The top and bottom borders of the box represent the 75%ile and 25%ile of $k$ values, respectively. The top and bottom whiskers represent the upper and lower limit of $k$ values in the groups. In addition, the smaller dots represent the outliers.

As a result of the higher SARS-CoV-2 RNA decay rates than norovirus within temperatures ranging between 3 and 37 °C, the sensitivity to COVID-19 back-estimation via WBE (Table 5) was greater than that of norovirus (Table S3) [10], considering both temporal temperature change and wastewater dilution using freshwater. The implications of in-sewer decay should be analyzed in association with specific pathogen species. In addition to the comparison between enveloped and nonenveloped viruses, RNA viruses (i.e., noroviruses, coronaviruses) and DNA viruses (i.e., adenoviruses) should also be differentiated. Furthermore, the difference between viral pathogens and bacterial ones might be even more distinguished [10].

*3.6. Implications, Limitations, and Future Perspectives*

The decay experiment is time-consuming; hence, the number of decay rates generated from an individual study is limited by resources. Through a systematic literature review and meta-analysis, this research synthesized experimental data from multiple studies and assessed the parameterization of in-sewer coronavirus decay for WBE applications. According to the sensitivity analysis of SARS-CoV-2 RNA decay, in-sewer decay in high temperature conditions should be considered for WBE back-estimation. Thus, recording the wastewater temperature at collection is highly recommended. Moreover, wastewater dilution in sewers induced by stormwater inflow may enhance the survivability of viable coronaviruses and result in an increased risk of infection through wastewater. This should be underlined particularly when the rainfall events are accompanied with cold weather, which is also conducive to the sustaining of virus infectivity. Preventive measures should be taken by the public health sectors to lower such risks.

Unavoidably, this research also has a few limitations regarding the scarcity of collected literature data in seawater, the absence of wastewater dilution experiments by other water types for coronaviruses, and a failure to include the impact of other sewer conditions. More available experimental data would make the statistical analysis consolidated. Further experimental results in wastewater mixed with fresh/saline water at various ratios can help quantify the actual dilution effect on coronavirus decay rates. The mathematical relationship of wastewater dilution at various gradients and the resulting changes in virus decay rates could be utilized for correcting WBE back-estimations. Revealing relevant mechanisms of coronavirus decay is also important for an in-depth understanding of virus fate in wastewater and requires further detailed studies using lab experiments. These outcomes could be extrapolated to other viral species as well.

Furthermore, it is essential to delineate the impact of various other in-sewer processes on coronavirus decay. Most importantly, the decay measured in bulk wastewater (or by in-vial tests) could significantly underestimate the actual decay in sewers [49], largely due to the virus adsorption to sewer biofilms. This has been shown by our sewer reactor studies not only for the enhanced decay of coronavirus infectivity [49], but also for SARS-CoV-2 RNA [50]. Another study [51] also reported the accumulation of SARS-CoV-2 RNA in sewer biofilms to 700 genome copies/cm$^2$, which demonstrated SARS-CoV-2 retardation in sewers. In addition, the sewer biofilms and sediments (particularly in gravity sewers) may shelter the associated virus particles from degradation and extend their preservation in sewers [52,53]. Later, the attached viruses might then be released into wastewater with biofilm sloughing when subjected to strong stormwater inflow. Given the complexity of a real sewer environment, the sensitivity analysis of WBE back-estimation to SARS-CoV-2 decay in this study is likely limited by the current knowledge gap.

In addition, some of the in-sewer chemical dosing strategies (oxygen, ferric salts, nitrate/nitrite, or free nitrous acid, etc.) may also disinfect microbes and accelerate the degradation of virus RNA at the same time [54]. This might be one of the reasons why there was no infectious SARS-CoV-2 isolated from the influent raw sewage samples and no reported COVID-19 transmission through wastewater exposure to date [55,56]. In-depth knowledge on the effect of the chemicals used in sewers and WWTPs on the survival of coronavirus could help develop effective disinfection strategies which might be applied to other environmental settings rather than wastewater only. Overall, understanding the fate of coronavirus after excretion from human bodies, including its in-sewer decay, is critical to ensure environmental and public health. The actual contribution of in-sewer SARS-CoV-2 RNA decay to overall variance in WBE back-estimation remains unclear due to the research gap in the unrevealed impact of in-sewer processes. To reduce these uncertainties in WBE application, more studies on the unknown in-sewer decay of coronavirus under different conditions are highly encouraged.

## 4. Conclusions

This paper analyzed the decay rate constants of coronaviruses in three types of water systematically collected from published literature. The impacts of wastewater temperature and dilution in sewers on virus decay and WBE back-estimation were further analyzed and discussed based on the collation of decay rate constants. Moreover, the decay of coronavirus was also compared with those of other viruses in wastewater through a literature survey to understand the differences among species. Accordingly, four major conclusions could be drawn as follows:

- SARS-CoV-2 RNA, as the biomarker for WBE investigations, has much lower decay rates and was less influenced by wastewater temperature variations than viable coronaviruses. However, higher wastewater temperature in summer or tropical regions could still increase the sensitivity of WBE back-calculations considerably from 0.06 to 0.18, resulting in a two-times higher relative variance in back-estimation of COVID-19 cases;
- Wastewater dilution by stormwater inflow might alleviate the decay of coronavirus infectivity. Cold weather along with heavy rainfall events and urban floods could further increase the risk of environmental transmission by improving virus survivability in wastewater;
- As an enveloped RNA virus, coronavirus experienced more rapid decay in wastewater than nonenveloped viruses such as norovirus, which led to the increased sensitivity of WBE back-estimation to in-sewer decay and, consequently, to a greater need to incorporate the decay rate as a correction factor;
- There is a lack of studies on coronavirus decay in saltwater, unlike some other, more extensively studied, waterborne enteric pathogens. Salt groundwater intrusion or the municipal usage of seawater (i.e., toilet flushing) in sewers might enhance the decay of coronavirus infectivity during wastewater transportation to wastewater treatment plants (WWTPs). However, this hypothesis needs further supporting experimental data.

**Supplementary Materials:** The following supporting information can be downloaded at: https://www.mdpi.com/article/10.3390/w15061051/s1, Table S1: Decay rates of infectious coronaviruses; Table S2: Decay rates of SARS-CoV-2 RNA; Table S3: Sensitivity ratio of predicted cases ($P_{infection}$) to the norovirus decay rate constants under different conditions assuming residence time in sewers as 12 h; Figure S1: Fitted curves using Arrhenius equation for infectious coronaviruses and SARS-CoV-2 RNA in wastewater and freshwater.

**Author Contributions:** Conceptualization, Y.G. and G.J.; formal analysis, Y.G.; data curation, Y.G.; writing—original draft preparation, Y.G.; writing—review and editing, Y.L., X.Z., S.G. and M.S.; visualization, Y.G.; supervision, G.J.; project administration, G.J.; funding acquisition, G.J. All authors have read and agreed to the published version of the manuscript.

**Funding:** This research was funded by the ARC Discovery Project, grant number DP190100385. Ying Guo receives a Ph.D. scholarship from the same project.

**Data Availability Statement:** Data is available in the Supplementary Materials.

**Conflicts of Interest:** The authors declare no conflict of interest. The funders had no role in the design of the study; in the collection, analyses, or interpretation of data; in the writing of the manuscript; or in the decision to publish the results.

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
