# Peer review of "Effects of Temperature and Water Types on the Decay of Coronavirus: A Review"

_water, doi:10.3390/w15061051_

Round 1

Reviewer 1 Report

Comments:

0. Major revision. 1. The novelty of this study should be inserted in the text clearly. 2. The advantages and disadvantages of this study should be investigated. 3. The stability of the coronavirus in wastewater should be presented. 4. The mechanism of the decay of coronavirus should be studied in detail. 5. The “introduction” section of the manuscript for wastewater treatment methods can be strengthened and supported with some papers related to the literature and cited (optional for authors): Journal of Alloys and Compounds 509 (2011), 4754-4764; Journal of Applied Polymer Science 109 (2008), 4043-4048; Journal of Environmental Health Science and Engineering 12 (2014), 96; Colloids and Surfaces B: Biointerfaces 76 (2010), 397-403; Applied Surface Science 255 (2009), 4171-4176; Colloids and Surfaces A: Physicochemical and Engineering Aspects 355 (2010), 183-186; Journal of Chemical & Engineering Data 55 (2010), 4660-4668; Journal of Environmental Chemical Engineering 7 (2019), 103243; Fibers and Polymers 11 (2010), 234-240; Journal of Molecular Catalysis A: Chemical 366 (2013), 254-260; Journal of Applied Polymer Science 122 (2011), 1489-1499; Dyes and Pigments 76 (2008), 684-689; Journal of the Taiwan Institute of Chemical Engineers 45 (2014), 2008-2020.      

Reviewer 2 Report

The work is interesting and following modifications are needed in the ms

The title could be only

 Effects of temperature and wastewater dilutions on the decay of coronavirus.

Line 16, 18, 45 etc. first full form then abbreviations (like you did at line 17), plz follow this rule in the entire ms

Line 28: mention at what temp

Appropriate literature survey is an integral part of a research which need to be done in  Line 45-64: This is the main part for leading the main objective of he study. So I encourage the authors to place  more literature in this section. Some of the highly relevant literatures are

2022. Paital, B, Das K, Malekdar F, Sandoval MA, Niaragh EK, Frontistis Z, Behera TR, Balacco G, Sangkham S, Hati A, Mousazadeh M. 2022. A State-of-the-Art Review on SARS-CoV-2 Virus Removal Using Different Wastewater Treatment Strategies. Environments 2022,  9(110), 1-27. https://doi.org/10.3390/environments9090110

2021. Mousazadeh M, Ashoori R, Paital B, Kabdaşlı I, Frontistis Z, Hashemi M, Sandoval M.A,  Sherchan S, Das K. 2021. Wastewater Based Epidemiology Perspective as a Faster Protocol for Detecting Coronavirus RNA in Human Populations: A Review with Specific Reference to SARS-CoV-2 Virus. Pathogens. 10, 1008. https://doi.org/10.3390/pathogens10081008,

Line 57-64: When you talk about sea water then salinity makes an important point to influence the osmotic pressure and thus degradation of RNA. So, can you correlate it?

Materials method section is ok but since it is literature survey based study, the same must be mention in title, abstract, intro, discussion and conclusion as well.

Why only  RT-PCR results were considered.

Table: This result is very important as RNA are very very sensitive to temp. So the possible solid causes behind such less change must b emphatically discussed with relevant citation.

Line 187, 192, 202: What was the salinity? This factor is very important to maintain the osmotic pressure for decay of virus. You must have the water, now you can measure the salinity and correlate the data.

Fig. 2 is blank?

Section 4: Make is concise to strictly half a paragraph, with a model figure drawn out of the study.

Rest part may be moved to discussion section.

Refernces

Few of the relevant references including suggested are missing.   

Reviewer 3 Report

This study reviewed the relationship between temperature and wastewater dilutions on coronavirus decay and its implications for wastewater-based epidemiology. In general, the topic is relatively well written and has a potential interest to the journal's readers. However, prior to giving the manuscript more consideration, a few issues need to be resolved.

·       All the abbreviations should be defined when used the first time before their applications elsewhere in the manuscript. The authors should also think of limiting abbreviations in the abstract.

·       The authors should also think of a proper selection of keywords. Some general keywords should be avoided.

·       The temperature and wastewater dilution concepts should be well highlighted in the introduction section.

·       The quality of the figures should be improved.

·       From the title, it appears that the dilutions' tangible effects on the decomposition of the coronavirus will be shown at the end of the day. But, given the nature of the investigation, that is far from being accomplished. The findings only draw attention to possible effects rather than actual ones. The authors ought to check and verify that.

Round 2

Reviewer 1 Report

Accept

Reviewer 2 Report

The ms nos may be accepted for publication.